# Recent Progress in the Study of Taste Characteristics and the Nutrition and Health Properties of Organic Acids in Foods

**DOI:** 10.3390/foods11213408

**Published:** 2022-10-28

**Authors:** Yige Shi, Dandan Pu, Xuewei Zhou, Yuyu Zhang

**Affiliations:** 1Food Laboratory of Zhongyuan, Beijing Technology and Business University, Beijing 100048, China; 2Key Laboratory of Flavor Science of China Gengeral Chamber of Commerce, Beijing Technology and Business University, Beijing 100048, China

**Keywords:** organic acids, taste presentation mechanism, dichotomous taste, nutritional health

## Abstract

Organic acids could improve the food flavor, maintain the nutritional value, and extend the shelf life of food. This review summarizes the detection methods and concentrations of organic acids in different foods, as well as their taste characteristics and nutritional properties. The composition of organic acids varies in different food. Fruits and vegetables often contain citric acid, creatine is a unique organic acid found in meat, fermented foods have a high content of acetic acid, and seasonings have a wide range of organic acids. Determination of the organic acid contents among different food matrices allows us to monitor the sensory properties, origin identification, and quality control of foods, and further provides a basis for food formulation design. The taste characteristics and the acid taste perception mechanisms of organic acids have made some progress, and binary taste interaction is the key method to decode multiple taste perception. Real food and solution models elucidated that the organic acid has an asymmetric interaction effect on the other four basic taste attributes. In addition, in terms of nutrition and health, organic acids can provide energy and metabolism regulation to protect the human immune and myocardial systems. Moreover, it also exhibited bacterial inhibition by disrupting the internal balance of bacteria and inhibiting enzyme activity. It is of great significance to clarify the synergistic dose-effect relationship between organic acids and other taste sensations and further promote the application of organic acids in food salt reduction.

## 1. Basic Introduction to Organic Acids

Organic acids refer to organic compounds that are acidic and contain one or more carboxyl groups. The most common organic acids are carboxylic acids (R-COOH), whose acidity originates from the carboxyl group (-COOH), except for organic acids such as sulfonic acid (R-SO_3_H), sulfinic acid (R-SOOH), and sulfuric acid (R-SH). Organic acids can be classified according to the differences in the number of carboxyl groups, hydroxyl groups, and carbon–carbon double bonds in their molecular structure: (1) aliphatic, alicyclic, aromatic, and heterocyclic acids, such as benzoic acid; (2) saturated or unsaturated acids, such as acetic acid and acrylic acid; (3) the number of carboxyl groups and whether the carboxyl groups are substituted, e.g., acetic acid (one carboxyl group), malic acid (two carboxyl groups), and citric acid (three carboxyl groups). Amino acids are organic compounds containing basic amino and acidic carboxyl groups (R-CHNH_2_-COOH), which are formed when the hydrogen atom on the carbon atom of a carboxylic acid is replaced by an amino group [1]. Amino acids are not included in this review.

Organic acids play an important role in maintaining the nutritional value and sensory quality of foods and are also an important class of food additives, including their use as preservatives, acidity regulators, antioxidants, etc., with a wide range of applications [2]. Among them, propionic acid and benzoic acid, when used as preservatives, can prevent food spoilage, effectively extending the shelf life; citric acid, malic acid, fumaric acid, and tartaric acid, when used as acidity regulators, can maintain or change the pH of food; and ascorbic acid is a common antioxidant that can prevent or delay the deterioration or oxidative decomposition of fats and oils or food components, and can improve food stability. The release of organic acids in the mouth also affects flavor perception during food oral processing due to the cross-modal relationship between aroma and taste [3]. According to the World Health Organization (WHO), globally, hypertension is the leading risk factor for mortality, and among Chinese residents, it is a chronic disease with high prevalence [2]. Therefore, it is especially important to achieve “salt reduction without reducing the salty taste” without affecting the flavor quality of food. When compounded with other tastes, the acidity produced by organic acids can have the effect of enhancing or inhibiting other tastes. A recent study found that malic acid, when added to low-sodium salts, reduced the bitterness introduced by the addition of potassium salts and improved the salty taste [4]. In terms of nutritional functions, organic acids are a class of nutrients that have demonstrated significant health effects such as anti-inflammatory properties, prevention of osteoporosis, regulation of host immune function, intestinal hormone production, anti-obesity, inflammation regulation, promotion of calcium absorption, and inhibition of platelet aggregation [5]. This review summarizes the content of organic acids in various types of foods, their taste characteristics, their interaction with other taste sensations, and the nutritional and health properties of some organic acids, with the aim of providing a clearer understanding of organic acids in food as well as potential scientific applications for organic acids in foods and for the development of healthy and tasty foods.

## 2. Content of Organic Acids in Food and Their Detection Methods

Seven methods are available to detect organic acids in food (Table 1): Ultraviolet (UV) spectrophotometry, Nuclear magnetic resonance (NMR), capillary electrophoresis, thin-layer chromatography, gas chromatography (GC), and liquid chromatography (LC). The above analytical methods can be used in tandem with a variety of detectors to achieve accurate identification. UV spectrophotometry is based on the formation of colored complexes of organic acids with chromogenic agents, binding coenzyme (e.g., reduced nicotinamide adenine dinucleotide or reduced nicotinamide adenine dinucleotide phosphate, etc.) reactions [6], and measuring them at specific wavelengths. Organic acids bind to coenzymes with high specificity, which can be used to detect isomers such as L/D-malic acid. To avoid interference, the initial separation of organic acids before detection is usually carried out by precipitation and ion exchange resin, which is a tedious process, and only one organic acid can be determined at a time [7]. The principle of NMR is that when the magnetic moment of the nucleus of the substance to be measured is not zero, it spins under the action of the external magnetic field and undergoes Newman splitting, resonantly absorbing a certain frequency of RF radiation to produce an increase in the energy level. The capillary electrophoresis method takes advantage of the differences in the charged nature, particle shape, and size of different substances and therefore the direction of movement and the speed of movement in the electric field for separation [8]. The principle of thin-layer chromatography for the detection of organic acids is based on the separation of different components by using Differences in the magnitude of the adsorption force of adsorbents on different components and the desorption effect of unfolders, which is quantified by the photosensitometry of each point on the plate [9]. Ion exchange chromatography uses ion exchange-affinity differences to achieve separation. Gas chromatography mainly uses the differences in boiling points, polarities, and adsorption properties of substances to achieve the separation of mixtures and is suitable for the determination of volatile organic acids, such as aromatic acids. However, this method is tedious because some organic acids are not volatile and need to be derivatized and then analyzed by gas chromatography. Liquid chromatography includes high-performance liquid chromatography, inverse high-performance liquid chromatography, and ultra-high-performance liquid chromatography. At present, high-performance liquid chromatography is the most commonly used method to determine organic acids in foods; it makes use of the difference in polarity of the various substances as well as the difference in adsorption with a stationary phase and a mobile phase creating separation, and, thus, leaving of the column at different times. Different peak signals are obtained through the detector, and finally, the substances contained in the substances to be measured are determined by analyzing and comparing these signals. It has several characteristics including a wide detection range, high sensitivity, good separation effect and short detection time.

Common types of organic acids detected in foods include citric acid, malic acid, fumaric acid, tartaric acid, maleic acid, quinic acid, aconitic acid, succinic acid, pyruvic acid, pyroglutamic acid, gallic acid, ascorbic acid, formic acid, acetic acid, propionic acid, butyric acid, lactic acid, creatine, and oxalic acid, which form an important class of taste substances. Organic acids not only regulate flavor but also participate in plant and animal metabolism and have nutritional value. The following subsection summarizes the types and contents of organic acids in common foods (Figure 1), aiming to provide a reference for future research on organic acids in foods.

**Figure 1 foods-11-03408-f001:**
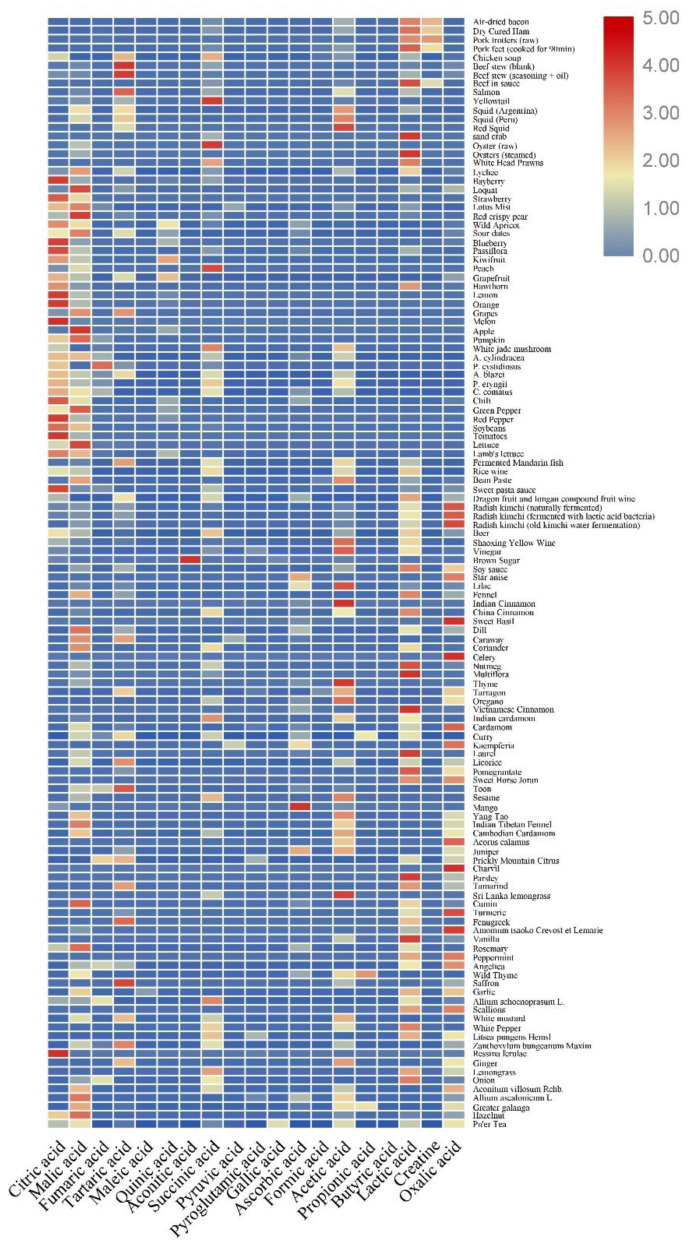
Heatmap analysis results of 19 major organic acids among 126 food samples [10,11,12,13,14,15,16,17,18,19,20,21,22,23,24,25,26,27,28,29,30,31,32,33,34,35,36,37,38,39,40,41,42,43,44,45,46,47,48,49,50,51,52]. The concentration result of organic acids (Appendix A) was conducted by Tbtools software (v0.665) [53], the standardized color intensity scale spans from the highest (dark red) to the lowest (dark blue) indicating the relative contents of organic acid content in foods from high to low.

**Table 1 foods-11-03408-t001:** Detection methods of organic acids in food.

Number	Separation Method	Detectors	Characteristic	References
1	GC	Hydrogen flame ion detector	Its sensitivity is high and the optimized response varies very little with factors such as detector temperature, flow rates of hydrogen and new carrier gas, and other	[54]
		Mass Spectrometry Detectors	High performance, high selectivity, high sensitivity, fast analysis speed, application range, can be used in conjunction with analysis	[55]
2	LC	Conductivity detector	It is sensitive and selective, and can be performed quickly and easily.	[56]
		Chemiluminescent detectors	The wavelength is used to identify substances with high selectivity and detection limits in the range of low micrograms to low picograms.	[57]
		Volt-ampere detector	Reproducibility with the simple and rapid procedure without derivatization of analytes.	[58]
		Mass Spectrometry Detectors	It is more sensitive, selective, and specific, and can determine trace amounts of substances.	[59]
		Fourier infrared detector	The coupling of LC and IR systems can be performed offline or in-line. On-line analysis offers the advantages of the high chromatographic resolution, real-time measurements, instrument simplicity, low cost, and the ability to use non-volatile buffers	[60]
		evaporative light scattering detector	It is more affordable than mass spectrometry detection methods and is also compatible with a wide range of solvents and gradient elutions.	[10]
		Refractive index detector	Simplicity of operation, cost, energy consumption, and availability in most QC labs are better than mass spectrometry detectors	[61,62]
		Diode array detector	The technique allows an easy quantification	[11,63,64]
3	Thin-layer chromatography	UV detector	Fast, sensitive (small amount), highly selective, simple, easy color development	[12]
4	Ion exchange chromatography	Conductivity detector	High precision and accuracy with good reproducibility	[13]
5	Capillary zone electrophoresis method	UV detector	High resolution, high automation, simple operation, fast speed, low chemical consumption, low sample preparation volume	[8]
6	NMR	Ultra-low temperature probe	Non-destructive, non-selective analysis, can directly analyze and detect a large number of compounds at once, but it is more difficult to detect complex mixtures	[13,18]
		Inverse detection probe		
7	UV spectrophotometer method	Photomultiplier tube	Reaction of organic acids with other substances and measurement of complexes at specific wavelengths	[6]

### 2.1. Content of Organic Acids in Fruits

Fruits commonly contain citric acid and malic acid. Among fruits, apples, loquats, lychees, crisp pears, and lotus fruits have the highest content of malic acid, i.e., 1493–10,950 mg/kg, and are classified as malic acid fruits [12,14,15,19,36,45,52]. Fruits such as lemon, orange, strawberry, prune, blueberry, and hawthorn have the highest citric acid content, i.e., 5635–51,490 mg/kg, and are classified as citric acid fruits [12,14,15,26,33,36,52]. There are also tartaric-acid-type fruits, with grapes containing 10,470 mg/kg of tartaric acid [27,52]. Furthermore, acetic acid, lactic acid, ascorbic acid, and oxalic acid are commonly found in fruits. For example, apples contain acetic acid (34 mg/kg) and oxalic acid (19 mg/kg) [45]; and lychee contains lactic acid (2050 mg/kg) and acetic acid (840 mg/kg) [19].

All stages of fruit development require large amounts of energy, which is mainly provided by the metabolism of different types of carbon-containing compounds, such as organic acids, amino acids, and sugars [52]. During respiration, a portion of the organic acids is consumed [65], mainly in the tricarboxylic acid cycle (TCA) and via glycolytic metabolism, producing acetate, pyruvate, and additional lactate. Pyruvate is converted to acetyl coenzyme A (CoA) under aerobic conditions and enters the TCA cycle [66]. The TCA cycle is an intermediate in the energy conversion process in plants, and the accumulation of its products is influenced by plant tissues, developmental stages, and environmental factors. Citrate and malate are both thought to function as signal transduction pathways in plants. A recent study found that the gene encoding P3B-ATPase (designated MdMa13) is a determinant of apple fruit acidity and that overexpression of MdMa13 in the apple calyx, tomato, or apple resulted in an increase in the malic acid content [67]. Attributes such as the nutritional and sensory quality and the palatability of fruits also depend on the balance of organic acids and sugars [68]. These two types of metabolites are directly related to central carbon metabolism and are also involved in the biosynthetic pathways of various compounds, such as amino acids, vitamins, and terpene component synthesis [52], which affect fruit aroma. Organic acids can enhance the characteristic flavor of fruit beverages. Panelists could identify the blackcurrant and orange-flavored solutions more accurately when citric or malic acid was added into the flavored solutions prepared with 1000 g of distilled water, 75 g pure cane sugar, and of concentrated fruit flavors (orange flavoring, 0.8 mL; blackcurrant flavoring, 1 mL) [69]. The ratio of sugar to acid in citrus juice contributes to the hedonic scores of sweetness and acidity and results in the overall acceptability of juice when increased from 12:1 to 22:1 [70]. Therefore, the optimization of the concentration ratio between the sugar and acid can improve the flavor intensity of the food beverage.

### 2.2. Content of Organic Acids in Vegetables

Organic acids, being one group of the primary metabolites of plants, are influenced by the growing region, climate, and species. Citric acid (770–10,640 mg/kg), malic acid (810–1900 mg/kg), and quinic acid (27–320 mg/kg) are the main acids of “fruity” vegetables, while the content of other organic acids is low [71]. The concentration of organic acids in peppers depends on the stage and part of ripening. The citric acid content in the peel of the pepper with the lowest maturity was 70,020–80,060 mg/kg, the malic acid content was 14,430 mg/kg, and the quinic acid content was 8960–12,720 mg/kg, while the citric acid content in the placenta was 98,220–98,930 mg/kg, the malic acid content was 13,650–45,960 mg/kg, and the quinic acid content was 5620–14,420 mg/kg [72]. Organic acids indirectly affect phenolic metabolism by changing the pH and are precursors to phenolic and flavor compounds. Pyruvic acid is one of the flavor-precursor substances that has been shown to be associated with the pungency of vegetables such as onions [73]. Among all onion samples, citric acid (485 mg/kg) and malic acid (436 mg/kg) were the main organic acids, followed by tartaric acid (188 mg/kg), oxalic acid (113 mg/kg), pyruvic acid (35.1 mg/kg), and fumaric acid (2.4 mg/kg). Leafy vegetables contain high concentrations of malic acid (2330–5750 mg/kg) and citric acid (1180–3560 mg/kg). Examples include lettuce and lamb’s lettuce [71]. In a study on different species of mushrooms, the main organic acid in the stalk was lactic acid (63,650 mg/kg), followed by malic acid (18,440 mg/kg) and citric acid (11,610 mg/kg), while succinic acid (4720 mg/kg), citric acid (12,090 mg/kg), and malic acid (34,890 mg/kg) were the main organic acids in the cap of the mushroom [3]. Citric acid was the highest organic acid in *P. matsutake* (237,810 mg/kg), and fumaric acid (96,110 mg/kg) was the main organic acid in *P. cystidiosus* mushrooms. Interestingly, mushrooms with a high total organic acid content also had higher equivalentumami taste concentrations [23].

### 2.3. Organic Acid Content in Livestock and Poultry Meat

Zhang et al. [44] studied the main organic acids in chicken broth samples stewed from different parts of Hailan brown chicken (including whole chicken, whole chicken legs, thigh meat, chicken breast meat, and chicken skeleton) and found they were citric acid (1531–4200 mg/kg), succinic acid (13,048–45,565 mg/kg), and tartaric acid (11,047–88,084 mg/kg), which accounted for more than 90% of the total organic acid content. Formic acid, acetic acid, citric acid, and fumaric acid are commonly used as feed additives for chickens, and the diversification of chicken feed is thought to help increase the content of organic acids such as tartaric acid and pyroglutamic acid [74]. The contents of lactic acid (3850–9050 mg/kg) and creatine (1480–3040 mg/kg) in ham are higher than those of other organic acids such as succinic acid and fumaric acid [30]. During pork stewing, the total organic acid concentration was reported to decrease significantly with increasing stewing time from 6800 mg/kg to 2370 mg/kg; the lactic acid content decreased (from 3490 mg/kg to 1320 mg/kg), the creatine content decreased (from 3180 mg/kg to 650 mg/kg), and the creatinine level increased (from 100 mg/kg to 2040 mg/kg) [13]. The reason for this is that under heated conditions, one molecule of water is eliminated from creatine, forming a ring structure that undergoes non-enzymatic conversion to creatinine [75]. Creatine plays an important role in muscle energy metabolism and is involved in energy transport processes in myocytes, which contributes to the maintenance of muscle strength [76]. Succinic acid reduces lipid oxidation and the premature browning of meat patties, while it increases the pH of minced meat and increases the reducing activity of methyl myoglobin, thus increasing the redness of meat [41]. The organic acid content of beef stew soup is generally low, containing citric acid (0.45 mg/kg), malic acid (1.488 mg/kg), tartaric acid (19.8 mg/kg), succinic acid (3.18 mg/kg), and lactic acid (1.26 mg/kg) [41]. Beef in sauce contains lactic acid (2250–4330 mg/kg), acetic acid (100–180 mg/kg), succinic acid (180–300 mg/kg), and creatine (890–1170 mg/kg). The lactic acid content of halal-sourced beef was reported to be slightly higher than that of non-halal beef (210 mg/kg) because the halal slaughtered beef process differs from that of normal slaughtered beef in that the pH of halal and non-halal beef decreases at different rates [43].

### 2.4. Organic Acid Content in Aquatic and Seafood Products

Succinic acid and its sodium salt are the main metabolites of the muscles of fish and shellfish and are some of the most important substances for the umami taste of seafood. It was determined that the succinic acid content in the yellow croaker was 52.6 mg/kg, with a taste activity value (TAV) of 0.50, and the tartaric acid content was 12.9 mg/kg, with a TAV of 0.86; both of these organic acids contribute significantly to the taste of yellow croaker, whereas citric acid (1.8 mg/kg), malic acid (6.6 mg/kg), and acetic acid (2.8 mg/kg) contribute less [25]. White and red muscle aqueous extracts of tropical wild tuna contain mainly lactic acid (2880–7020 mg/kg) and creatine (1311.3–27,973.29 mg/kg). It has been found that most of the creatine is phosphorylated in the resting muscles and provides energy for muscle contraction in the form of high-energy phosphate [76]. During exercise, glucose catabolism and oxidation in the muscle of tuna produce pyruvate, which produces lactate faster than the body can metabolize it, resulting in a 720-fold increase in the lactate concentration [77]. The organic acid content in the flesh of different species of the squid varies, which may be related to adenosine triphosphate (ATP) coenzyme-related activity or action causing changes in organic acids. A comparison of the organic acid content and TAVs of three squid species revealed that tartaric acid (865.661 mg/kg, TAV = 57.71), acetic acid (1334.325 mg/kg, TAV = 12.58), and malic acid (835.881 mg/kg, TAV = 1.68) contributed significantly to the taste of Argentine squid, while tartaric acid (639.272–643.102 mg/kg, TAV = 42.61–42.87) and acetic acid (957.847–2249.096 mg/kg, TAV = 9.036–21.21) were the main taste substances of Peruvian and red squid [21]. The lactic and succinic acid contents in the heads of South American white shrimp were revealed to be 15.27 mg/kg and 13.32 mg/kg, respectively [17], and sand crabs contained lactic acid (436.2 mg/kg) and succinic acid (87.7 mg/kg) [31], similar to the results of previous studies where only lactic acid and succinic acid were detected in different parts of different species of crabs. Succinic acid (97–123.2 mg/kg) is the main acidic component responsible for the taste of raw and autoclaved cooked oysters; No lactic acid was detected before the steaming of oysters, while the lactic acid content was 258.3 mg/kg after steaming, with TAV greater than 1, which is the main acidic component determining the taste of steamed oysters, but the succinic acid content became 0 after steaming [48]. Similarly, in studies related to several species of fish, it was found that different modes of steaming and boiling had different effects on the organic acid content. Although the organic acid content of both steamed and boiled fish decreased, the organic acid content of boiled fish was lower than that of steamed fish [78]. This change and differences in organic acid contents due to different heating methods should be further investigated.

### 2.5. Content of Organic Acids in Fermented Foods

Several metabolic pathways are activated in fermented foods during fermentation (Figure 2), such as the amino acid catabolic pathway (Ehrlich pathway), sugar metabolic synthesis pathway (Harris pathway), glycolysis, lactic acid fermentation, and acetate fermentation [74,79,80]. As a result, lactic acid and acetic acid, as the products of lactic acid fermentation, acetic acid fermentation, and glucose metabolism, are high in fermented foods. The organic acid content is related to the fermentation substrate, fermentation time, and fermentation method. The high microbial diversity at the beginning of fermentation and the acidic anaerobic fermentation environment may affect the diversity and abundance of species, leading to large changes in transcript levels during fermentation [81].

The oxalic acid content in radish kimchi under different fermentation methods (natural fermentation, lactic acid bacterial fermentation, and old kimchi water fermentation) decreased (302,297–104,762 mg/kg) as the fermentation time increased due to oxalic acid-generating esters and alcohols during fermentation [49]. However, the contents of lactic acid (694–7938 mg/kg) and acetic acid (192–665 mg/kg) showed an increasing trend, and the contents of these acids in lactic acid bacterial fermentation were significantly higher than those in the other two fermentation methods. The tartaric acid content showed a decrease followed by an increasing trend, whereas succinic acid showed the opposite trend, and the citric acid content showed differences due to the fermentation methods. Xiao et al. analyzed the organic acid content and its changes during the fermentation of kimchi in Sichuan using genetic and metabolic profiles and found that lactic acid, acetic acid, and malic acid had important effects on the sour taste of kimchi [82]. During the fermentation of *Siniperca odorata*, the contents of succinic, acetic, lactic, malic, and tartaric acids increased gradually and were positively correlated with the fermentation time [23]. Some of the organic acids may come from the fish itself, while most of them came from the fermentation process. The content of organic acids in traditional Shaoxing Huangjiu of different ages, except pyruvic acid, was reported to decrease annually [40]. Succinic acid contributes to a large extent to the taste of low-vintage Huangjiu, with a sour and spicy taste. Lactic and citric acids contribute more to the taste of traditional Shaoxing Huangjiu in higher vintages than other organic acids, resulting in a more harmonious Huangjiu taste. The contents of different organic acids in rice-flavored white wine showed different trends during fermentation [51]. The contents of acetic, lactic, and succinic acids increased in the fermentation mash on day 12 compared to the initial fermentation mash, and the total organic acid content increased throughout the fermentation process. The concentration detection of organic acid in fermented food has many positive effects on quality control and quality improvement. For example, a geographical classification model can be established to determine the source of white wine according to the different contents of organic acids and some trace elements [83]. At the same time, excess organic acid in wine resulting in a sour taste affects the consumers’ preference. Therefore, understanding the suitable content of organic acids in wine will help to produce more ideal wine products [84].

### 2.6. Content of Organic Acids in Seasonings

Organic acids in condiments come from the raw materials and process of condiments. The sources of organic acids in vinegar are diverse, most of which are produced by fermentation [82], and only trace amounts of organic acids originate from the raw materials. Acetic acid (26,926.71–54,215.31 mg/kg) and lactic acid (1547.31–17,185.09 mg/kg) account for more than 80% of the total organic acid content in vinegar, and its fermentation consists of three major biochemical reaction processes [5]: (1) starch is hydrolyzed into monosaccharides, oligosaccharides, and short-chain polysaccharides by enzymes secreted by microorganisms; (2) ethanol-fermenting yeast converts fermentable sugars into ethanol and CO_2_, accompanied by glycerol, fatty acids, succinic acid, etc.; and (3) ethanol is oxidized to acetic acid, which is further fermented by acetic acid bacteria. The TAV of citric acid (16.1–36.1 mg/kg) was found to be greater than 1, followed by lactic acid (10.79–139.7 mg/kg), succinic acid (54.63–124.1 mg/kg), and malic acid (12.69–48.22 mg/kg), which contributed more to the acidity of a sweet pasta sauce [55]. Among 18 strong spices, all of them contained oxalic acid except Indian cardamom. Thyme had the highest organic acid content (54,670 mg/kg), and anise had the lowest (11,590 mg/kg) [39]. Lactic acid was detected in most of the 29 light fragrances studied, followed by oxalic acid. The total organic acid content of the *Acorus calamus* (38,800 mg/kg) was significantly higher than that of the other spices. The second-highest organic acid content (30,050 mg/kg) was found in prickly shamrock [85]. Oxalic acid was detected in all of the 20 spices studied except white pepper. Wild mint had the highest total organic acid content (339,580 mg/kg), and onion had the highest succinic acid content (83,530 mg/kg) [86].

### 2.7. Summary

Statistical analysis of the organic acid content in 125 kinds of food or seasoning such as fruits, vegetables, meat products, seafood, and fermented products indicated the following: (1) fruits generally contain citric acid and malic acid, and most of them also contain acetic acid, lactic acid, ascorbic acid, and oxalic acid; (2) vegetables mostly contain citric acid, malic acid, and fumaric acid; (3) meat foods mostly contain succinic acid and lactic acid—in chicken soup and beef soup, citric acid, tartaric acid, and oxalic acid were also detected, and fumaric acid was only detected in some pork products; (4) the main organic acids in seafood are malic acid, tartaric acid, succinic acid, acetic acid, and lactic acid, and a small amount of citric acid can also be detected. Creatine is an organic acid unique to animal muscle, detected in poultry and livestock meat and fish; (5) fermented foods mostly contain citric acid, malic acid, tartaric acid, succinic acid, and lactic acid, with high levels of acetic acid; and (6) seasonings, the organic acids of which are more diverse, show no obvious pattern and commonly include malic acid, lactic acid, oxalic acid, tartaric acid, succinic acid, acetic acid, and ascorbic acid. Based on the detailed data on the organic acids in different foods, it is possible to predict the food sensory characteristics, monitor the fermentation process, and to identify the origin and quality of foods, providing a basis for future research on food development.

## 3. Flavoring Properties of Organic Acids

### 3.1. Taste Mechanism of Acidity

In 1898, William firstly demonstrated that the sour taste of acids was due to hydrogen ions [87]. However, in further studies, it was found that organic acids cause a sour taste, not only due to the H^+^ concentration, but also possibly due to the organic acid anions (acid ion) in the solution [88]. The chemical structure, dissociation constant and anion concentration of acid play a key role in the sour perception of organic acid. At the same concentration and pH value, increasing more carboxyl groups reduces the intensity of acidity. Besides, acetic acid (monocarboxylic acid) had a higher intensity of sour perception than citric acid (tricarboxylic acid); lactic acid (monocarboxylic acid) had a lower intensity of sour perception than malic acid (a dicarboxylic acid). These results showed that the molecular weight, type, and number of anions substituents affected the acidity intensity [89]. Enhancing the H^+^ binding to receptors by reducing the positive charge of the membrane; reacting with receptor sites or saliva, and weak acids maintain a normally constant pH by further dissociation is the main sour perception mechanisms of acidic anions [35]. Lawless suggested that some anions, such as lactate ions, inhibit the sour taste [12]. However, Neta et al. showed no significant relationship between the suppression of the organic acid sour taste in aqueous solutions with constant pH and the total concentration of organic anions [90].

In 2006, Huang et al. [91] found no taste response to sour taste stimuli by knocking out the PKD2L1 gene in mice and concluded that sour taste perception is dependent on the expression of PKD2L1. Later, Horio et al. [92] studied mice lacking the PKD2L1 and PKD1L3 genes and found that compared to mice without the knockout genes, the pulsatile tympanic nerve response to sour taste was reduced by 25–45%, along with a 25–45% reduction in the sour taste response in type III taste cells, suggesting that PKD2L1 contributes partially to the sour taste response in mice, but that receptors other than PKD are also involved in sour taste transmission. Patients with extraordinarily high citric acid detection thresholds lack acid-sensitive ion channels (ASICs) and PKD2L1, whereas normal human EuP cells with normal acid taste perception express ASICs and PKD2L1, further suggesting that the acid taste perception pathway is the result of multiple mechanisms interacting with each other [93]. Liman et al. [94,95] identified OTOP1 (apical ion channel through which H^+^ passes) as an essential transducer receptor supporting sour tastes. OTOP1 introduces H^+^ into the cell. H^+^ ions not only directly depolarize the taste cell but also directly alter the membrane potential (Vm), and the change in intracellular pH blocks the K^+^ channels (Kir2.1), thereby amplifying depolarization [96]. The resulting depolarization triggers voltage-gated Na^+^ channels [97,98], generating action potentials that activate voltage-gated Ca^2+^ channels and trigger the release of neurotransmitters in the synaptic vesicles [99], which are subsequently transmitted to the brain by the bulbar and glossopharyngeal nerves (Figure 3).

In addition to mammals, several fishes, birds, and amphibians have also been reported that could perceive sour taste [100], but the sour taste perception mechanisms of these animals are still unclear. Among the approximately 9900 species of birds on earth, only six species reported that they could perceive the sour taste, such as Agelaius phoeniceus, Sturnus vulgaris, and Nymphicus hollandicus [101]. Amphibian Rana catesbeiana also did not like the sour taste. Organic acid could also be perceived by some fishes, such as Carassius auratus, Cyprinus carpio, and Acipenser baerii; they rejected the food pellets flavored with citric acid [102]. The sour taste receptor recognition in Drosophila was related to the mammalian acid taste receptor, OTOP1. Knockout of the OtopLA gene greatly impaired acid perception in flies, making Drosophila unable to recognize acidity [103].

### 3.2. Binary Taste Interactions

Currently, the binary interaction analytical method has been wildly used in aroma-aroma, aroma-taste, and taste-taste interactions to investigate the synergistic and inhibitory effects between aroma and taste perception [104,105,106,107]. There were three levels of interactions when assessing mixed taste compounds [98]: (1) chemical interactions: altering taste intensity and generating new taste sensations; (2) oral physiological interactions: secondary interactions in which a compound in the mouth has the potential to interfere with taste receptors or taste transduction mechanisms associated with another compound; and (3) cognitive interactions, affecting taste signals transmitted to the central nervous system in the brain. Kroeze et al. [108] used the split tongue method to study taste transmission to neuronal interactions in the brain. Such peripheral interactions between taste compounds may occur on taste receptor cells or at multiple sites within cells. Interactions between taste sensations may also depend on psychological effects, and different psychological demands may lead people to reorder or switch taste sensations in different ways [109].

The human taste system is capable of recognizing five major tastes: sweet, sour, bitter, salty, and umami. When multiple different taste compounds are mixed, one or more of these taste sensations will show enhancement and inhibition effects. Studying the perceptual and physiological outcomes of taste interactions is the key to understanding complex taste perceptions and plays an important role in healthy food and drug development. Although three different taste interactions have been reported in the literature, binary taste interactions lack in-depth studies. By summarizing the interaction of sourness produced by organic acids with other tastes, it was found that in most cases, the results of the interaction varied with reagents and concentrations. Low concentrations of organic acids enhanced saltiness perception and reduced them at high concentrations. Increasing concentrations of organic acids reduced umami perception. At low concentrations, organic acids had variable effects on sweet perception, while at high concentrations, sour and sweet perception inhibited each other. Sour and bitterness enhanced each other at low concentrations; at moderate concentrations, bitterness was inhibited and sour was enhanced; and at high concentrations, sour was inhibited and had variable effects on bitterness. Currently, there are not enough perceptual data to draw firm conclusions ofz some interactions.

### 3.3. Interaction of Organic Acids with Saltness

Adding organic acids with sub-threshold values into the sodium chloride solutions could enhance its saltiness perception [110,111]. Keiko et al. [112] added vinegar at a concentration of half the detection threshold (0.77 mg/L) for each panelist to a salt solution and found that both the detection and recognition thresholds for salt were reduced. Ulla et al. [113] demonstrated that salt content in solid foods such as rye bread could be significantly reduced by the addition of organic acids without reducing saltiness and mouthfeel. This finding was further validated in a recent study on taste perception in the oral processing of white bread, where the addition of 400–1200 mg/L malic acid enhanced saltiness perception [114]. Hsueh et al. [115] confirmed that chitinous nanofibers with citric acid (3000 mg/L) and malic acid (4000 mg/L) could improve the saltiness of tilapia fillets cured in 40,000 mg/L NaCl solution. Ko et al. found that vinegar significantly increased the saltiness of soups with low concentrations of added salt. Salt substitutes, high-pressure treatments, and organic acid mixtures were effective methods for salt-reducing without a decrease of saltiness perception on cooked ham [116]. Low-sodium salt was invented to address the problem of the high salt content in food. Most of the low-sodium salts currently available on the market were potassium and calcium as partial replacements for sodium. Wang et al. proposed that when L-malic acid, succinic acid, citric acid, and fumaric acid were added to low-sodium salt a significant saltiness enhancement effect on low-sodium salt was obtained [4].

According to Keast et al. [117], organic acids and Na^+^ may inhibit the sour taste through physiological interactions at the cellular/epithelial level or cognitive interactions in the cerebral cortex. One of the transduction mechanisms of sour taste perception has now been postulated to be an amiloride-sensitive Na^+^ channel [118,119], and this Na^+^ channel has been shown to be a salty taste perception-related channel, so sour–salty taste may share the same ion channel [120]. Furthermore, a recent neuroimaging study showed that the human cerebral cortex had spatially distinct but overlapping cortical activations for each taste quality, with umami taste in the anterior insula overlapping with the effects of salt and sour stimuli [121].

### 3.4. Interaction of Organic Acids with Umami

The perception intensity of umami gradually decreased with increasing citric acid concentration when different concentrations of citric acid (0.24–6.00 mg/L) were added into monosodium glutamate (8.46 mg/L) [122]. It is possible that the sour masks the umami [123]. The temporal dominance of sensations (TDS) was a popular method to analyze the binary combination of umami and sourness, and it was found that the coexistence of lactic acid and Monosodium glutamate (MSG) shortened the duration of umami, while MSG had no inhibitory effect on the duration of lactic acid [124], indicating that the interaction between sourness and umami is asymmetrical.

### 3.5. Interaction of Organic Acids with Sweetness

It is generally accepted that the effect of organic acids on sweetness is variable at low concentrations [125]. Rose concluded that citric acid enhanced the sweetness of sucrose at concentrations of 4990–17,000 mg/L in the concentration range of 77–730 mg/L [126]. Citric acid inhibited sweetness at different concentrations only when the sucrose concentration was greater than 205,200 mg/L [127]. In contrast, Junge et al. concluded that the addition of citric acid or tartaric acid to aqueous solutions always inhibited sucrose sweetness [128]. Presumably, the inhibitory effect of sweetness on sourness may be due to competitive inhibition. When type II cells secrete ATP in response to sweet taste stimuli, some will bind to the P2Y4 purinoceptors of type III cells causing an indirect response, resulting in reduced sour taste perception [129,130].

### 3.6. Interaction of Organic Acids with Bitterness

Acids enhance bitterness at low concentrations (77–730 mg/L); at moderate concentrations (near threshold levels) bitterness is suppressed and acidity is enhanced; and at high concentrations (greater than the threshold), acidity is suppressed but the effect on bitterness is variable (enhanced or diminished) [117,126]. Robinson [131] argued that because few foods are intensely bitter, there is a tendency to interpret strongly acidic flavors as bitter, and bitter and sour flavors may be confused. In food applications, organic acids have been shown to partially mask or inhibit the bitter taste of calcium chloride. Similarly, Wang et al. [4] indicated that various organic acids (L-malic, citric, fumaric, and succinic acids) had significant bitterness-masking effects on low sodium salts with added potassium chloride. Other studies have shown that citric acid can inhibit the bitter taste of drugs in solution. Sotoyama et al. [132] mixed different concentrations of citric acid with oral-disintegrating tablets of olopatadine for sensory evaluation and confirmed that citric acid could inhibit the bitter taste of the drugs during the oral disintegration of tablets. The suppressed bitterness may be attributed to a masking effect. Calcium imaging analysis of HEK293T cells heterologously expressing the bitter taste receptor (TAS2R16) showed that organic acids inhibit the bitter taste receptor and that this inhibition depends on the pH generated by the added acid, not on the concentration [133].

## 4. Physiological Properties of Organic Acids

### 4.1. Provide Energy

The TCA cycle is a unified cyclic process of metabolism in an organism, consisting of a series of biochemical reactions in the mitochondrial matrix (Figure 4); is a common pathway for the complete oxidation and release of energy from sugars, lipids, and proteins (amino acids); and is a hub for their interconnection and transformation [134,135,136]. During sustained exercise, the concentration of L-malate increases significantly, which enhances the malate–aspartate shuttle, promotes ATP synthesis and utilization in the mitochondria, and improves respiratory efficiency [137]. Thus, L-malate has a significant effect on the recovery of physical strength after physical exertion. Exogenous L-malate is easily absorbed, and supplementation further increases the activity of malate dehydrogenase and mitochondrial malate dehydrogenase in cells, maintains higher TCA cycle intermediates, increases TCA and the malate–aspartate shuttle rate, meets the energy requirements of the body under special conditions, reduces the level of serum creatine kinase during exercise, reduces damage to skeletal muscle, and significantly improves exercise capacity [138]. Succinate is oxidized to fumarate in the TCA cycle via succinate dehydrogenase and functions as a key intermediate in the mitochondrial electron transport chain complex, thus providing a link between the TCA cycle and the mitochondrial electron transport chain [139]. Creatine and phosphate provide energy through the creatine kinase (CK) and PCr systems [140]. CK enzymatically cleaves ATP to ADP, generating free energy that supplements the energy required for the resynthesis of ADP to ATP, thereby enhancing exercise capacity [141]. Meanwhile, creatine enters the cytoplasm via the creatine transporter (CRTR) to maintain glycolytic ATP levels.

### 4.2. Regulation of Metabolism

The immunomodulatory and physiological functions of lactate act through the regulation of the cellular cytokine network and the signaling systems of the intestinal mucosal system [142]. Metabolic intermediates of the TCA cycle have recently been shown to play an important role in the regulation of innate immune cell responses. It was reported that L-malic acid also reduces lipid peroxidation in aged rat tissue (liver and heart). Malic acid was used directly as a precursor to producing succinate by the reduction of ferredoxic acid, while generating ATP, which protected myocardial cell membrane integrity [143], and was used clinically as a component of the cardiac basal fluid. Zhou et al. [144] found that excess citric acid promoted melanogenesis in mouse cells but inhibited melanogenesis in human cells by regulating tyrosinase activity. Acetic acid and propionic acid increased satiety and delayed gastric emptying time, thereby decreasing the postprandial glucose response and insulin response [145]. Carbohydrates are most easily digested under alkaline conditions, and acidity leads to impaired absorption. In vitro experimental data demonstrated that a pH below 4.0 with the addition of acetic acid inactivated α-amylase [146] and reduced its release until the nutrients reached the small intestine—the channel responsible for 30–40% of complex carbohydrate digestion. Serum cholesterol and triglycerides were reduced by 0.3% acetic acid in rats on a high-cholesterol diet [147], promoting lipid homeostasis and hypocholesterolemic effects in vivo. Creatine supplementation also helped to lower cholesterol and triglycerides [148], control lipid levels, reduce fat accumulation in the liver, and lower cysteine levels, thus reducing the risk of heart disease, among other risks [149].

### 4.3. Bacterial Inhibition

Citric acid and its salts have been shown to be effective in controlling pathogens in poultry. Recent studies have also shown that citric acid can be used as an antitumor agent [150]. Citric acid is known to inhibit bacteria and enhance the inhibition of pathogens through chelation. Creatine, lactic acid, and acetic acid also have corresponding functions. Undissociated organic acid molecules can enter intracellularly through the bacterial cell membrane by free diffusion [151] where they dissociate to produce acid ions (ROO^−^) and protons (H^+^) [152]. The dissociated protons and acid ions cannot cross the membrane by free diffusion, which causes H^+^ to accumulate in the bacterial cell, resulting in a decrease in pH. Bacteria need to maintain a neutral pH environment to ensure the function of enzymes [153], thus forcing bacteria to consume ATP to release intracellular hydrogen ions to restore their pH balance, which competes with normal bacterial growth and metabolism for energy and reduces bacterial productivity [154]. Enrichment of acid ions in bacteria alters their environment, interfering with or even blocking DNA synthesis in the nucleus, thus inhibiting bacterial division and proliferation. Acid ions also allow for differential transcription of RNA, leading to altered metabolic pathways. Furthermore, it also leads to the blockage of intracellular protein synthesis, from being able to inhibit the normal growth of bacteria (Figure 5).

## 5. Conclusions and Outlook

With the use of advanced separation and detection technologies, we can detect more types of organic acids in food. Organic acids provide basic energy for plants and animals, and they influence food flavor as taste substances and aroma precursors. We concluded that the organic acid content in foods decreased by different heating methods (stewing, autoclaving, steaming). Foods with strong umami characteristics (chicken soup, mushrooms, seafood, etc.) had higher organic acid concentrations (total organic acid content higher than 10,000 mg/kg). The mechanism of acid taste perception and conduction is mainly due to the binding of type III taste receptor cells with sour taste substances, which are transmitted to the relevant taste cortex of the brain through the bulbar and glossopharyngeal nerves. However, the interactions between organic acids and other taste sensations are complex and variable, and the mechanisms need to be further investigated. The flavor-enhancing effect of organic acids deserves further study. Furthermore, in nutritional health, organic acids also provide energy to the body, regulate metabolism in order to reduce cardiovascular diseases, and demonstrate physiological activities such as the inhibition of bacteria.

## Figures and Tables

**Figure 2 foods-11-03408-f002:**
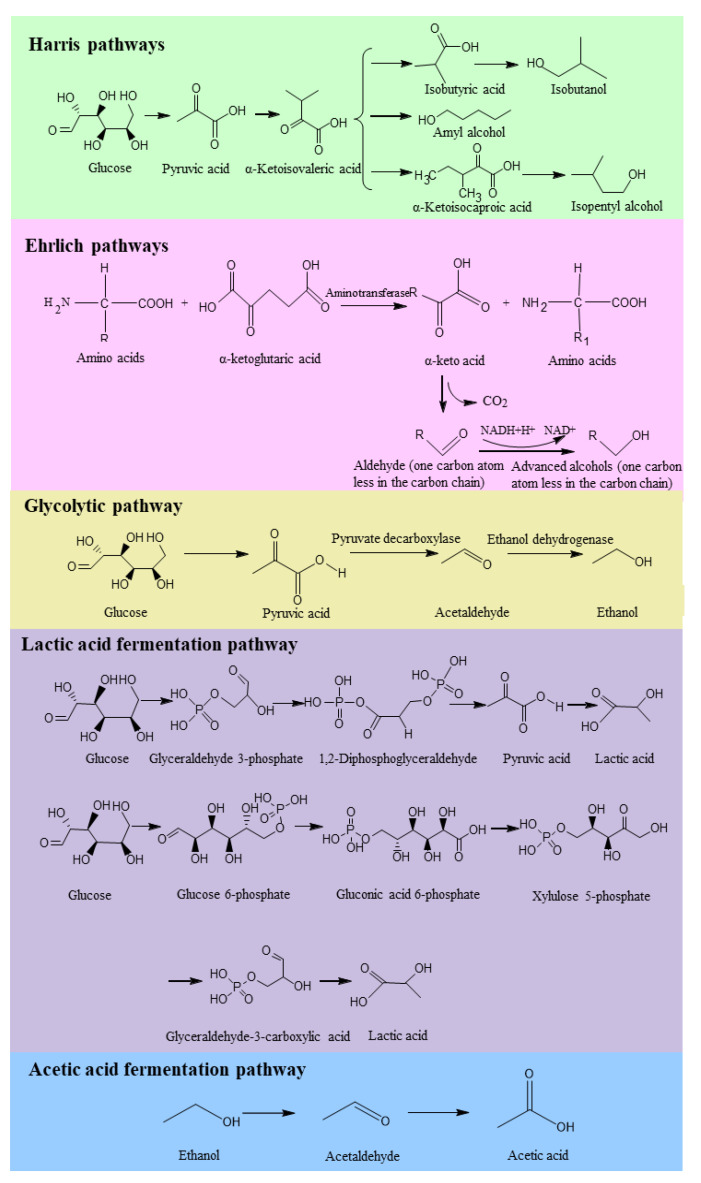
Multiple metabolic pathways in fermented foods during the fermentation process [80].

**Figure 3 foods-11-03408-f003:**
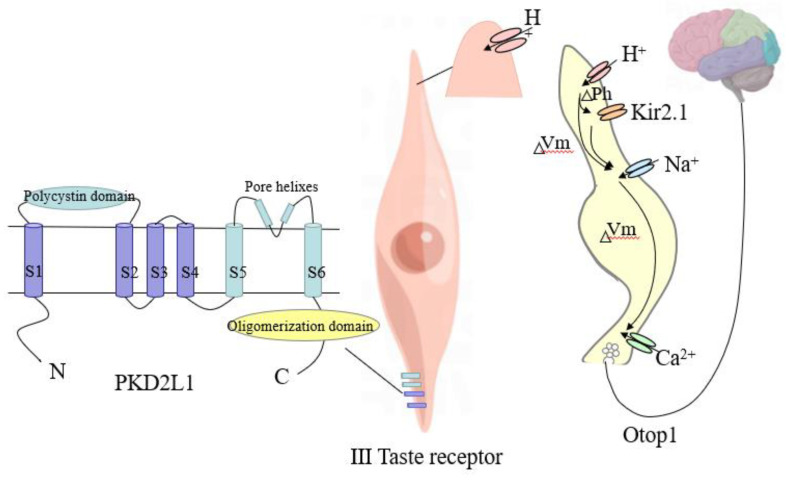
Type III sour taste receptor apical OTOP1 channel and PKD2L1 gene [96].

**Figure 4 foods-11-03408-f004:**
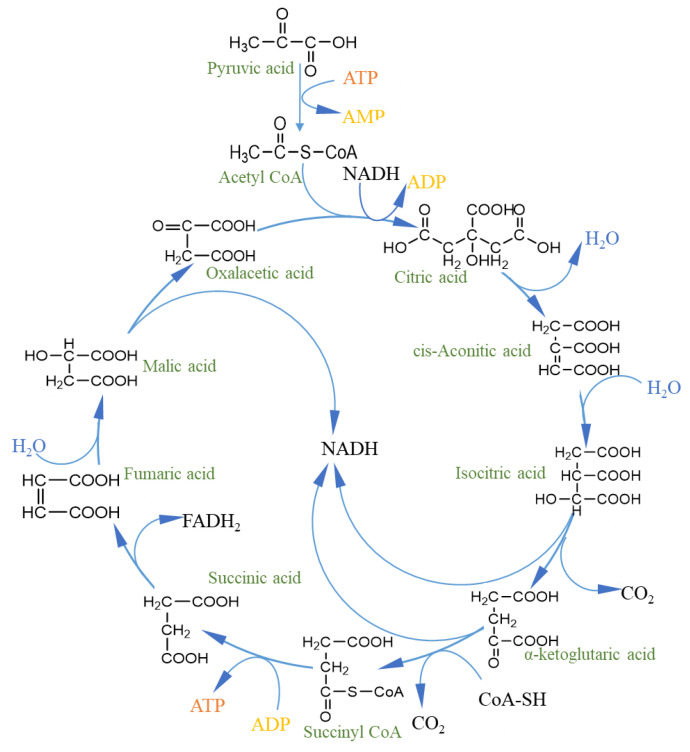
The tricarboxylic acid cycle provides energy [134,135,136].

**Figure 5 foods-11-03408-f005:**
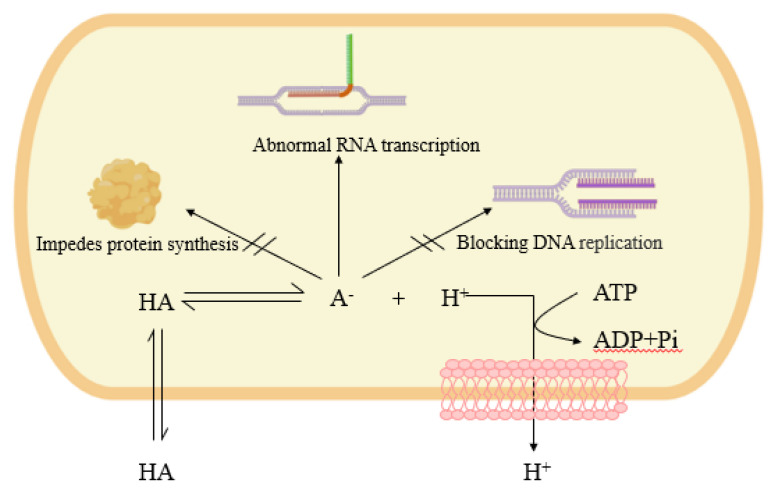
Mechanisms of bacterial inhibition by organic acids [150,151,152,153,154].

## Data Availability

Not applicable.

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
