# Peer review of "Recent Progress in the Study of Taste Characteristics and the Nutrition and Health Properties of Organic Acids in Foods"

_foods, 2022, doi:10.3390/foods11213408_

Round 1

Reviewer 1 Report

Despite this work is a review, the aim must be defined better; please, try to reformulate the abstract focusing on the aim.

Please, change all Kg with kg, according to the International System of Units

---------

Aim of this article is to be reviewing the amount of organic acids in different food matrices underlining their   importance as natural components of fresh foods, their micro nutritional values, their influence in enhancing or reducing the perception of saltiness, sweetness and of umami flavor; again, their antibacterial activity has been reported. The work is rather well organized, even if not particularly interesting considering, for example the long lists of acid organics cited in the test. The abstract have to be rewritten, focusing, if possible, on the positive effect of maintaining the original levels of organic acids in food matrices and on the future perspectives. The Authors have to pay attention at the unit: e.g change all Kg in kg, according to the  International System of Units

Author Response

Dear editors and reviewers,

Thank you for your and the reviewers’ comments concerning our manuscript. Those comments are very valuable and helpful for revising and improving our manuscript. We read all the comments carefully and made revisions which we hope meet with approval. The point-by-point responses to the reviewers’ comments are appended below. The detailed revisions with “Tracked Changes” are in the manuscript, and most of long sentences deleted or revised are marked red for better review. 

Thank you for your patience once more. We are looking forward to your positive response.

Yours sincerely,
Yige She

Reviewer 1: Aim of this article is to be reviewing the amount of organic acids in different food matrices underlining their   importance as natural components of fresh foods, their micro nutritional values, their influence in enhancing or reducing the perception of saltiness, sweetness and of umami flavor; again, their antibacterial activity has been reported. The work is rather well organized, even if not particularly interesting considering, for example the long lists of acid organics cited in the test. The abstract have to be rewritten, focusing, if possible, on the positive effect of maintaining the original levels of organic acids in food matrices and on the future perspectives. The Authors have to pay attention at the unit: e.g change all Kg in kg, according to the  International System of Units.

Response: Thanks for your useful comment. The manuscript was carefully revised according to your suggestion. 

Q1.the long lists of acid organics cited in the test.

Response: Thanks. The purpose of heat map (Fig. 1) is to compare the types and contents of organic acids in different foods, so that we can better understand the characteristics of organic acid content in food matrix. These results are beneficial to the study of the concentration setting of organic acids and different taste effects, and have important reference significance for the adjustment and setting of the formula in the follow-up food.

Q2.The abstract have to be rewritten, focusing, if possible, on the positive effect of maintaining the original levels of organic acids in food matrices and on the future perspectives.

Response: Thanks. Abstract section has been rewritten, the organic acid content in some foods has been removed, and the positive role and future application of studying the organic acid content in food matrix have been increased. The modified summary is as follows: “Organic acids, could improve the food flavor, maintain the nutritional value, and extend the shelf life of food. This review summarizes the detection methods and concentrations of organic acids in different foods, as well as their taste characteristics and nutritional properties. The composition of organic acids is varies in different food.Fruits and vegetables often contain citric acid, creatine is a unique organic acid found in meat, fermented foods have a high content of acetic acid, and sea-sonings have a wide range of organic acids. Determination of the organic acid contents among different food matrices allows us monitoring the sensory properties, origins identification and quality control of foods, and further provides a basis for food formulation design. The taste characteristics and the acid taste perception mechanisms of organic acids have made some pro-gress, and binary taste interaction is the key method to decode the multiple taste perception. Real food and solution models elucidated that the organic acid has an asymmetric interaction effect on the other four basic taste attributes. In addition, in terms of nutrition and health, organic acids can provide energy and metabolism regulation to protect human immune and myocardial systems. Moreover, it also exhibited bacterial inhibition by disrupting the internal balance of bacteria and inhibiting the enzyme activity. It is of great significance to clarify the synergistic dose effect rela-tionship between organic acids and other taste sensations and further promote the application of organic acids in food salt reduction.”.

The positive effect of maintaining the original levels of organic acids in food matrices and on the future perspectives were added: “Organic acids can enhance the characteristic flavor of fruit beverages. Panelists could identify the blackcurrant and orange-flavored solutions more accurately when citric or malic acid were added into the flavored solutions prepared by 1000 g of distilled water, 75 g pure cane sugar and of concentrated fruit flavors (orange flavoring, 0.8 mL; black-currant flavoring, 1 mL.) [69]. The ratio of sugar to acid in citrus juice contributing to the hedonic scores of sweetness, acidity and resulting in overall acceptability of juice when increased from 12:1 to 22:1 [70]. Therefore, optimization of the concentration ratio be-tween the sugar and acid can improve the flavor intensity of the food beverage.” Line 156-164. “The concentration detection of organic acid in fermented food has many positive ef-fects on quality control and quality improvement. For example, a geographical classifi-cation model can be established to determine the source of white wine according to the different contents of organic acids and some trace elements [83]. At the same time, ex-cess organic acid in wine resulting in sour taste affects the consumers' preference. Therefore, understanding the suitable content of organic acids in wine will help to produce more ideal wine products [84].” also has been added. Line 291-297.

Four references were added (Line 155-162 and Line 288-295):

  1. Zampini, M., Wantling, E., Phillips, N., Spence, C. Multisensory flavor perception: Assessing the influence of fruit acids and color cues on the perception of fruit-flavored beverages. Food quality and preference, 2008, 19, 335-343.
  2. Yu, Y. S., Xiao, G. S., Xu, Y. J., Wu, J. j., Fu, M. q., Wen, J. Slight fermentation with Lactobacillus fermentium improves the taste (sugar: acid ratio) of citrus (Citrus reticulata cv. chachiensis) juice. Journal of Food Science, 2015, 80, 2543-2547.
  3. Zhang, J., Tian, Z. Q., Ma, Y. Q., Shao, F. L., Huang, J. L., Wu, H., Tian, L. Origin identification of the sauce-flavor Chinese Baijiu by organic acids, trace elements, and the stable carbon isotope ratio. Journal of Food Quality,2019, 2019.
  4. Chidi, B. S., Bauer, F. F., Rossouw, D. Organic acid metabolism and the impact of fermentation practices on wine acidity: A review. South African Journal of Enology and Viticulture, 2018, 39, 1-15.

Q3. The Authors have to pay attention at the unit: e.g change all Kg in kg, according to the  International System of Units.

Response: Thanks. The full text unit has been modified, and Kg is changed to kg according to the International System of Units.

Other revisions:

  1. We adjusted the author information, firstly, Xuewei Zhou was added for edition and writing of our revision manuscript. Secondly, we deleted the “author equal to the contribution to this work” co-author Dandan Pu was revised to the first corresponding author. Therefore, the author order of this work was listed as follows: Yige Shi, Dandan Pu*, Xuewei Zhou and Yuyu Zhang*. The adjustment of author information changes were agreed by all authors, and their signatures were listed in Table S1.

Table S1. Author agreement of author information changes

Author name

Signature

Yige Shi

Dandan Pu

Xuewei Zhou

Yuyu Zhang

  1. “The release of organic acids in the mouth also affects the aroma perception during food oral processing due to the cross-modal relationship between aroma and taste. ”was add Line 50-52.
  2. One reference “Pu, D. D., Shan, Y. M., Wang, J., Sun, B. G., Xu, Y. Q., Zhang, W. G., Zhang, Y. Y. Recent trends in aroma release and perception during food oral processing: A review. Critical Reviews in Food Science and Nutrition, 2022, 1-17.”was added.Line 52.
  3. “(Figure 1)” was added. Line 114.
  4. “Capillary electrophoresis method ” was revised to“Capillary zone electrophoresis  method”. Table 1.
  5. “It was also found that citric acid was an important organic acid in all eight immature soybean species, with an average concentration of 2,831 mg/kKg, followed by malic acid (2,106 mg/kg) [32].” has been deleted. Line 191-193.
  6. “2.3. Organic acid content in meat (livestock, poultry)” was revised to“2.3. Organic acid content in livestock and poultry’s meat”. Line 194.
  7. “This” was deleted. Line 238.
  8. “Organic acids in condiments come from the raw materials and process of condiments.” was added. Line 303.
  9. “Based on the detailed data of the organic acids in different foods, it is possible to predict the food sensory characteristics, to monitor the fermentation process and to identify the origin and quality of foods, providing a basis for future research on food development.” was added.Line 339-342.
  10. “first” was revised to“firstly”. Line 345.
  11. (Figure 3)” was added.Line 381.
  12. At present, the binary interaction analytical method have been wild used in the aroma-aroma, aroma-taste, taste-taste interactions to investigate the synergistic and inhibitory effects between the aroma and taste perception [103-106].” was added. Line 395-397.
  13. Four references “Pu, D. D., Shan, Y. M., Zhang, L. L., Sun, B. G., Zhang, Y. Y. Identification and Inhibition of the Key Off-Odorants in Duck Broth by Means of the Sensomics Approach and Binary Odor Mixture. Journal of Agricultural and Food Chemistry, 2022, 70, 13367–13378.” and “Yang, Y. N., Yu, P., Sun, J. Y., Jia, Y. M., Wan, C. Y., Zhou, Q., Huang, F. H. Investigation of volatile thiol contributions to rapeseed oil by odor active value measurement and perceptual interactions. Food Chemistry, 2022, 373, 131607.” and “Liang L., Duan W., Zhang J. C., Huang Y., Zhang Y. Y., Sun B. G. Characterization and molecular docking study of taste peptides from chicken soup by sensory analysis combined with nano-LC-Q-TOF-MS/MS. Food Chemistry, 2022, 383, 132455.” and “Liang L., Zhou C. C., Zhang J. C., Huang Y., Zhao J., Sun B. G., Zhang Y. Y. Characteristics of Umami Peptides Identified from Porcine Bone Soup and Molecular Docking to the Taste Receptor T1R1/T1R3. Food Chemistry, 2022, 387, 132870.” were added. Line 397.
  14. “(Figure 4)” was added. Line 493.
  15. “(Figure 5)” was added. Line 554.
  16. “We found that many foods with higher levels of umami contain higher concentrations of organic acids, and the flavor-enhancing effect of organic acids deserves further study. The content of organic acids in foods decreases after heating, and the reasons for the decrease need to be further explored.”was revised to “We concluded that the organic acid content in foods decreased by different heating methods (stewing, autoclaving, steaming). Foods with strong umami characteristic (chicken soup, mushrooms, seafood, etc.) had the higher organic acid concentrations (total organic acid content higher than 10000 mg/kg).”. Line 560-564.
  17. “The flavor-enhancing effect of organic acids deserves further study.”was add Line 568-569.

Reviewer 2 Report

Manuscript is very informative and well-written overview that will be useful to many researchers. I do not have any significant comments on the text, except for a few wishes:

1. Authors should pay more attention to the common misconception that the sour taste of an acid solution is determined by the concentration of hydrogen ions (pH). The text talks about that, but very briefly.

2. The review uses data obtained on humans and other mammals. I would like to see in the text, perhaps in the final part, how specific or universal are the considered sensory properties of organic acids? For example, would these properties hold true for fish or amphibians?

3. I have a lot of comments about Figure 1. Many notations in this figure are incomprehensible and need to be explained. For example, what do the dendrograms at the top and right side of the figure mean? What do the numbers below mean (lowest row)? What do the numbers above the left colored panel mean? Both color panels (left and right) have similar colors, which is confusing. Maybe just keep the right panel, but transform it so that in each of its color block present a gradual transition from pale to intense tone? Undoubtedly, figure #1 needs to be modified.

Author Response

Dear editors and reviewers,

Thank you for your and the reviewers’ comments concerning our manuscript. Those comments are very valuable and helpful for revising and improving our manuscript. We read all the comments carefully and made revisions which we hope meet with approval. The point-by-point responses to the reviewers’ comments are appended below. The detailed revisions with “Tracked Changes” are in the manuscript, and most of long sentences deleted or revised are marked red for better review. 

Thank you for your patience once more. We are looking forward to your positive response.

Yours sincerely,
Yige She

Reviewer 2: Manuscript is very informative and well-written overview that will be useful to many researchers. I do not have any significant comments on the text, except for a few wishes:

Q1. Authors should pay more attention to the common misconception that the sour taste of an acid solution is determined by the concentration of hydrogen ions (pH). The text talks about that, but very briefly.

Response: Thank you. We have revised the manuscript in detail according to your useful suggestions. The structure of organic acid has been supplemented to further affect the sour taste. The sentences “The chemical structure, dissociation constant and anion concentration of acid play the key role in sour perception of organic acid. At the same concentration and pH value, increasing more carboxyl groups reduces the intensity of acidity. Besides, acetic acid (mono carboxylic acid) had the higher intensity of sour perception than citric acid (tri carboxylic acid); lactic acid (mono carboxylic acid) had the lower intensity of sour per-ception than malic acid (a dicarboxylic acid). These results showed that the molecular weight, type and number of anions substituents affected the acidity intensity [89]. En-hancing the H+ binding to receptors by reducing the positive charge of the membrane; reacting with receptor sites or saliva, and weak acids maintain a normally constant pH by further dissociation is the main sour perception mechanisms of acidic anions [46].” were added. Line 346-355.

One references were added (Line 355):

  1. CoSeteng, M. Y., McLellan, M. R., Downing, D. L. Influence of titratable acidity and pH on intensity of sourness of citric, malic, tartaric, lactic and acetic acids solutions and on the overall acceptability of imitation apple juice. Canadian Institute of Food Science and Technology Journal, 1989, 22, 46-51.

Q2. The review uses data obtained on humans and other mammals. I would like to see in the text, perhaps in the final part, how specific or universal are the considered sensory properties of organic acids? For example, would these properties hold true for fish or amphibians?

Response: Thank. In addition to mammals, other animals also perceive sour taste and describe its sensory characteristics. The detailed discussion were added: “In addition to mammals, several fishes, birds, and amphibians have also been re-ported that could perceive sour taste [99], but the sour taste perception mechanisms of these animals still unclear. Among the approximately 9900 species of birds on earth, only six species reported that they could perceive the sour taste, such as Agelaius phoeniceus, Sturnus vulgaris and Nymphicus hollandicus [100]. Amphibian Rana catesbeiana also didn't like sour taste. Organic acid could also be percieved by some fishes, such as Carassius auratus, Cyprinus carpio and Acipenser baerii, they rejected the food pellets flavoured with citric acid [101]. The sour taste receptor recognition in Drosophila was related to the mammalian acid taste receptor, OTOP1. Knockout of the OtopLA gene greatly impaired acid perception in flies, making Drosophila unable to recognize the acidity [102].”. Line 380-389.

Four references were added (Line 380-389):

  1. Frank, H. E., Amato, K., Trautwein, M., Maia, P., Liman, E. R., Nichols, L. M., Schwenk, K., Breslin, P. A., Dunn, R. R. The evolution of sour taste. Proceedings of the Royal Society B, 2022, 289, 20211918.
  2. Jetz, W., Thomas, G. H., Joy, J. B., Hartmann, K., Mooers, A. O. The global diversity of birds in space and time. Nature, 2012, 491, 444-448.
  3. Marui, T., Caprio, J. Teleost gustation. In Fish chemoreception , 1992, 6, 171-198.
  4. Ganguly, A., Chandel, A., Turner, H., Wang, S., Liman, E. R., Montell, C. Requirement for an Otopetrin-like protei D.n for acid taste in Drosophila. Proceedings of the National Academy of Sciences, 2021, 118, 2110641118.

Q3. I have a lot of comments about Figure 1. Many notations in this figure are incomprehensible and need to be explained. For example, what do the dendrograms at the top and right side of the figure mean? What do the numbers below mean (lowest row)? What do the numbers above the left colored panel mean? Both color panels (left and right) have similar colors, which is confusing. Maybe just keep the right panel, but transform it so that in each of its color block present a gradual transition from pale to intense tone? Undoubtedly, figure #1 needs to be modified.

Response: Figure 1 was conducted by Tbtools software[64], and it has been modified accordingly by removing the tree diagram and keeping only one legend. The standardized color intensity scale spans from the highest (dark red) to the lowest (dark blue), indicating the relative contents of organic acid content in foods from high to low. The numbers at the bottom of the figure are modified to the names of different foods. The original data of Figure 1 has been supplemented in the attached table.

One references were added (Line 115):

  1. Chen, C. J., Chen, H., Zhang, Y., Thomas, H. R., Frank, M. H., He, Y. H., Xia, R. TBtools: an integrative toolkit developed for interactive analyses of big biological data. Molecular plant, 2020, 13, 1194-1202.

Figure 1. Heatmap analysis results of 19 major organic acids among 125 food samples[17, 22, 24-63]. The result was conducted by Tbtools software(v0.665) [64], the standardized color intensity scale spans from the highest (dark red) to the lowest (dark blue), indicating the relative contents of organic acid content in foods from high to low.

Other revisions:

  1. We adjusted the author information, firstly, Xuewei Zhou was added for edition and writing of our revision manuscript. Secondly, we deleted the “author equal to the contribution to this work” co-author Dandan Pu was revised to the first corresponding author. Therefore, the author order of this work was listed as follows: Yige Shi, Dandan Pu*, Xuewei Zhou and Yuyu Zhang*. The adjustment of author information changes were agreed by all authors, and their signatures were listed in Table S1.

Table S1. Author agreement of author information changes

Author name

Signature

Yige Shi

Dandan Pu

Xuewei Zhou

Yuyu Zhang

  1. “The release of organic acids in the mouth also affects the aroma perception during food oral processing due to the cross-modal relationship between aroma and taste. ”was add Line 50-52.
  2. One reference “Pu, D. D., Shan, Y. M., Wang, J., Sun, B. G., Xu, Y. Q., Zhang, W. G., Zhang, Y. Y. Recent trends in aroma release and perception during food oral processing: A review. Critical Reviews in Food Science and Nutrition, 2022, 1-17.”was added.Line 52.
  3. “(Figure 1)” was added. Line 114.
  4. “Capillary electrophoresis method ” was revised to“Capillary zone electrophoresis  method”. Table 1.
  5. “It was also found that citric acid was an important organic acid in all eight immature soybean species, with an average concentration of 2,831 mg/kKg, followed by malic acid (2,106 mg/kg) [32].” has been deleted. Line 191-193.
  6. “2.3. Organic acid content in meat (livestock, poultry)” was revised to“2.3. Organic acid content in livestock and poultry’s meat”. Line 194.
  7. “This” was deleted. Line 238.
  8. “Organic acids in condiments come from the raw materials and process of condiments.” was added. Line 303.
  9. “Based on the detailed data of the organic acids in different foods, it is possible to predict the food sensory characteristics, to monitor the fermentation process and to identify the origin and quality of foods, providing a basis for future research on food development.” was added.Line 339-342.
  10. “first” was revised to“firstly”. Line 345.
  11. (Figure 3)” was added.Line 381.
  12. At present, the binary interaction analytical method have been wild used in the aroma-aroma, aroma-taste, taste-taste interactions to investigate the synergistic and inhibitory effects between the aroma and taste perception [103-106].” was added. Line 395-397.
  13. Four references “Pu, D. D., Shan, Y. M., Zhang, L. L., Sun, B. G., Zhang, Y. Y. Identification and Inhibition of the Key Off-Odorants in Duck Broth by Means of the Sensomics Approach and Binary Odor Mixture. Journal of Agricultural and Food Chemistry, 2022, 70, 13367–13378.” and “Yang, Y. N., Yu, P., Sun, J. Y., Jia, Y. M., Wan, C. Y., Zhou, Q., Huang, F. H. Investigation of volatile thiol contributions to rapeseed oil by odor active value measurement and perceptual interactions. Food Chemistry, 2022, 373, 131607.” and “Liang L., Duan W., Zhang J. C., Huang Y., Zhang Y. Y., Sun B. G. Characterization and molecular docking study of taste peptides from chicken soup by sensory analysis combined with nano-LC-Q-TOF-MS/MS. Food Chemistry, 2022, 383, 132455.” and “Liang L., Zhou C. C., Zhang J. C., Huang Y., Zhao J., Sun B. G., Zhang Y. Y. Characteristics of Umami Peptides Identified from Porcine Bone Soup and Molecular Docking to the Taste Receptor T1R1/T1R3. Food Chemistry, 2022, 387, 132870.” were added. Line 397.
  14. “(Figure 4)” was added. Line 493.
  15. “(Figure 5)” was added. Line 554.
  16. “We found that many foods with higher levels of umami contain higher concentrations of organic acids, and the flavor-enhancing effect of organic acids deserves further study. The content of organic acids in foods decreases after heating, and the reasons for the decrease need to be further explored.”was revised to “We concluded that the organic acid content in foods decreased by different heating methods (stewing, autoclaving, steaming). Foods with strong umami characteristic (chicken soup, mushrooms, seafood, etc.) had the higher organic acid concentrations (total organic acid content higher than 10000 mg/kg).”. Line 560-564.
  17. “The flavor-enhancing effect of organic acids deserves further study.”was add Line 568-569.

Round 2

Reviewer 1 Report

Dear All

thank You very much for improving Your work